# MAC/PHY Comprehensive Visible Light Communication Networks Simulation

**DOI:** 10.3390/s20216014

**Published:** 2020-10-23

**Authors:** Edmundo Torres-Zapata, Victor Guerra, Jose Rabadan, Martin Luna-Rivera, Rafael Perez-Jimenez

**Affiliations:** 1Institute for Technological Development and Innovation in Communications (IDeTIC), Universidad de Las Palmas de Gran Canaria, 35001 Las Palmas, Spain; etorres@idetic.eu (E.T.-Z.); jrabadan@idetic.eu (J.R.); rperez@idetic.eu (R.P.-J.); 2Facultad de Ciencias, Universidad Autonoma de San Luis Potosi (UASLP), San Luis Potosi 78295, Mexico; mlr@fciencias.uaslp.mx

**Keywords:** visible light communications, IEEE 802.15.7, channel simulation, network simulation, MAC layer simulation, MAC/PHY simulation

## Abstract

In this paper, the effect of channel conditions on the global behavior of a wireless Visible Light Communications (VLC) optical network are studied. It presents a system-level simulator that considers jointly a channel propagation model and the MAC mechanisms to have a realistic description of the network, even in situations where the emitted signal is heavily affected by reflections in any close surface or obstacle. The resulting platform also accurately evaluates both Line-Of-Sight (LOS) and Non-LOS (NLOS) contributions on each node and enables the effective use of Carrier Sense Multiple Access with Collision Avoidance (CSMA/CA) schemes as defined by IEEE 802.15.7r1 standard, as well as allows a correct evaluation of lifelike problems such as the effect of hidden nodes. This work shows the necessity of accurately modeling VLC MAC layer performances, taking also into account the physical nature of visible light propagation in indoor scenarios.

## 1. Introduction

The development of the Internet of Things (IoT) and wireless sensor networks is currently producing a significant impact on many industries, enabling the integration of valuable real-time information into different kinds of systems. These systems are usually based on Radio Frequency (RF) technologies, and their massive integration is progressively saturating the radio spectrum in commercial bands. Furthermore, some of these industries, concretely those that have ElectroMagnetic Compatibility (EMC) restrictions such as biomedical or nuclear facilities, may have severe difficulties in integrating wireless IoT solutions due to their strict regulations.

A strong candidate technology to provide a solution to both radio spectrum saturation and EMC constraints in some cases is Visible Light Communications (VLC). VLC is based on the use of Solid State Lighting (SSL) devices such as White Light Emitting Diode (WLED) lamps, and thanks to the moderate bandwidth of these commercial lighting lamps, this technology is currently capable of providing downlink speeds up to a few hundred Mbps [1] in indoor scenarios. VLC has experienced a steep growth since the first proofs of concept developed by Nakagawa et al. in the early 2000s [2], and there are currently two consolidated standards (IEEE 802.15.7r1 [3] and ITU G.9991 [4]) and some task groups working on Optical Wireless Communication (OWC).

Besides the insensitivity of OWC systems to RF interference, VLC is used as a dual system (illumination and communications). This leads to an important issue of this technology that sometimes is neglected, which is the physiological necessity to avoid flickering or any other perceivable effect on the illumination. This was analyzed in-depth during the definition of IEEE 802.15.7 standard, resulting in three main transmission modes which use On-Off Keying (OOK), Pulse Position Modulation (PPM), and Color Shift Keying (CSK). Furthermore, since VLC operates in a human-sensitive range of wavelengths, it is also necessary to carry out a multi-objective optimization when designing VLC facilities [5], since both data transmission coverage and lighting uniformity must be ensured.

Furthermore, VLC has been traditionally advertised as an inherently secure technology. Although it is more technically-complicated to compromise an OWC link than an RF-based system, light is not completely confined within a room if it has any type of potential leak (windows, door locks, etc.) [6].

The research on VLC has been focused during the last years on demonstrating the limits of the technology in terms of achievable bandwidth and channel modeling [7,8]. Nonetheless, the scientific community has not put sufficient effort into developing and simulating MAC-layer protocols taking into account the particularities of OWC links. Concretely, IEEE 802.15.7r1 MAC is essentially the same as IEEE 802.15.4 (used by ZigBee and Bluetooth), and it does not consider that user nodes may not have visibility among them, generating multiple collisions and hence, impairing the network performance.

Moreover, research on VLC network performance is generally based on either pure MAC-layer simulation [9] or pure physical channel simulation [10]. It is straightforward to demonstrate that the conclusions extracted from these works are strongly biased and cannot be considered as accurate models of real-world deployments. Moreover, there are few contributions in the literature addressing the problem in a holistic manner [11]. Due to the relative novelty of VLC there are not many software libraries for traditional network simulation platforms (OMNET++, NS-2, OPNET, Prowler, etc.). In this work, a comprehensive VLC network simulator that considers not only the MAC-layer definition of IEEE 802.15.7r1 standard, but also an accurate and dynamic channel impulse response estimation between nodes, is presented. The MAC-layer mechanisms have been developed for OMNET++, while the optical wireless channel impulse response is calculated using a Modified Monte Carlo Ray tracing (MMCRT) algorithm [12]. The resulting platform allows one to accurately evaluate the performance of VLC networks, taking into account both Line-Of-Sight (LOS) and Non-LOS (NLOS) contributions on each node. Depending on the characteristics of the scenario (materials, geometry, etc.), the user nodes may be aware (or not) about the transmissions of neighboring nodes. This enables the effective use of Carrier Sense Multiple Access with Collision Avoidance (CSMA/CA) that IEEE 802.15.7r1 defines. Nonetheless, as it will be shown during the results sections of this work, this effectiveness is lost depending on the distance between nodes and their optical front-end sensitivities, ultimately diminishing the network performance. Hence, this work provides scientific evidence about the necessity of accurately modeling VLC network taking into account both the logical processes within the MAC layer and the physical nature of visible light propagation in indoor scenarios.

The remainder of this paper is structured as follows. In Section 2, an in-depth analysis regarding the state-of-the-art contributions related to MAC layer simulation for VLC networks is presented. Section 3 presents a clinical description of the schemes and processes related to IEEE 802.15.7r1 MAC layer. Moreover, Section 4 introduces the structure of the proposed simulation platform and describes in detail each comprising part. The description of the experiments carried out in this work are presented in Section 5, and the results are illustrated in Section 6. Finally, Section 7 extracts and presents the conclusions of this work and provides insight into potential future research directions.

## 2. Related Work

In this section, the current state-of-the-art solutions on VLC network simulation are analyzed. This work is primarily focused on the IEEE 802.15.7 standard, since it is the most studied reference in the literature. Nonetheless, as mentioned in Section 1, ITU G.9991 standard on high-speed VLC transceivers is already published and there are other undergoing standards such as IEEE 802.15.13 [13] and IEEE 802.11bb [14]. However, these last two standards are not VLC-specific, but broader in terms of working wavelength.

It has been observed that papers addressing VLC MAC-layer simulation or system-level simulation are scarce in the literature. These simulations enable a deep comprehension of the network performance, allowing the detection of issues prior to the system’s physical implementation. Furthermore, network performance simulation can also be used to optimize MAC-layer configuration or even rapid prototyping novel protocols. In this analysis, the contributions to the aforementioned topic have been categorized depending on their consideration of the impact of physical phenomena. Some works address network simulation only from a MAC-layer viewpoint, whilst others include some physical layer parameters such as the Signal-to-Noise Ratio (SNR) on their schemes. However, no scheme combining physical and MAC simulation was found.

Nobar et al. analyzed the performance of the IEEE 802.15.7 CSMA/CA mechanism under saturated traffic conditions using Markov chains [15]. The authors focused primarily on obtaining a modeling tool rather than simulating actual VLC networks, since no real-world metrics taking into account the standard presented. Moreover, the model’s accuracy is limited to small network sizes and a significant convergence between the model and simulation is obtained above 12 users, which may be unlikely to occur in reality due to the reduced coverage area of VLC access points [16]. Some months later, Mehr, Nobar, and Niya carried out a similar work but for unsaturated traffic in Ref. [17]. The work was based on the same model, but some approximations made for saturated traffic were accommodated for the new conditions.

IEEE 802.15.7 standard considers only half-duplex transmission since communications are controlled by the central node using a superframe structure, as it will be described in Section 3. Wang et al. simulated a contention-based full-duplex communication mode, in which the access point (or central node) can also initiate data transmissions in the downlink [18]. The authors showed that the standard may benefit from this type of access in star topologies, significantly increasing the downlink throughput. Nonetheless, as it happened with Nobar et al., the physical layer was not considered, and their results must be carefully considered. In addition, a relevant number of authors have proposed the use of different working wavelengths for downlink and uplink [19,20]. Therefore, downlink could be carried out using the broadcast mode of the standard, whilst the uplink could still use the mechanisms used for star topology.

In Ref. [21], the impact of the hidden node problem during channel contention was studied. Due to the directivity of VLC endpoints, the emitter was not able to detect surrounding nodes outside of its Field Of View (FOV). This situation leads to a misuse of the channel, producing collisions. This work presented a comparative study of the metrics between a network in which there are no hidden nodes and a network in which nodes are unable to properly sense the channel. Nevertheless, the work did not consider the NLOS contribution caused by multipath propagation in the scenario, which may allow channel occupation detection in real scenarios.

Unlike the previous works, Dang and Mai included some physical parameters into an extended version of Dobar’s Markov chain model, resulting in a 3D graph structure [11]. Although the authors stated that a single reflection was considered to simulate its impact on the network performance, there is a lack of results in this regard. The authors did not present the resulting impulse responses in the scenario, despite their modeling effort. Therefore, their results are hardly reproducible. Nonetheless, Dang and Mai demonstrated that considering the optical channel may have a dramatic impact on parameters such as latency, throughput, and reliability.

Abdalbahi et al. developed a VLC simulation block for NS3 in ref. [22] and performed validation experiments. Their experimental results were accurate in respect to the simulation results, but the only tested scenario was point-to-point. Therefore, the impact of contention-based access using CSMA/CA, which is the most important part of the MAC-layer, was not properly validated. A few years later, Makvandi et al. developed real implementation of IEEE 802.15.7 and tested a four-node star topology [23]. Nonetheless, due to the link geometry and optical receiver sensitivity, CSMA/CA was not actually working, obtaining a significant number of collisions and hence impairing network performance.

Although it falls outside IEEE 802.15.7 standard, Orthogonal Frequency Division Multiplexing (OFDM) has been amply studied and experimentally evaluated in VLC [24], and the currently under-development IEEE 802.15.13 and IEEE 802.11bb standards are focused on using this modulation scheme due to its flexibility. There has been a significant effort on developing multiple access techniques based on OFDM in VLC [25,26]. Nonetheless, there is no available literature (up to the knowledge of the authors) involving MAC and PHY simulations considering specifically OFDM.

## 3. Mac Layer in IEEE 802.15.7 Standard

The IEEE 802.15.7 standard defines both the PHY and MAC layers for short-range optical wireless communications using visible light. The MAC sublayer is responsible for the channel access coordination of the user nodes (devices) in the network. The simulator developed in this work is based on this standard since it is the most extended and has a detailed description. Moreover, it is the most embraced by the scientific community and its information is open access.

Some of the tasks performed by the MAC layer are beacon management, channel access control, Guaranteed Time Slots (GTS) management, frame validation, acknowledge frame delivery (ACK), and association/disassociation of nodes [3].

The standard supports three network topologies. Peer-to-peer configuration is defined for establishing communication between two endpoints, broadcasting mode describes a one-to-all communication mode, and finally, star topology specifies a flexible bidirectional communication mode between user nodes and a central node. This work focuses on star-topology scenarios, where all the mechanisms from VLC standard’s MAC layer are needed. As it was briefly commented, this topology comprises of a coordinator that manages communication and multiple devices connected to it. The communication between a user and coordinator is carried out using a half-duplex VLC link. Since star-topology defines a centralized network, users cannot communicate directly between them. If they need to send a message to another user inside the network, they have to ask the coordinator to act as an intermediary. The following subsection shortly describe some key MAC procedures of IEEE 802.15.7.

### 3.1. Synchronization and Superframe Structure

IEEE 802.15.7 can operate using an optional Superframe mode to achieve low latency and facilitate synchronization to the user nodes. Figure 1 illustrates the Superframe structure of the IEEE 802.15.7 standard. The Superframe is the time distribution that starts when a dedicated coordinator sends a beacon frame in predetermined intervals, and ends when the next beacon arrives. This time difference is called Beacon Interval (BI). Besides the implicit time synchronization of the beacon frames, they also provide information about the channel distribution.

The Superframe is divided into Active and Inactive periods. During the Active period, the user nodes are allowed to communicate. Otherwise, in the Inactive period, they are in idle mode to prevent struggling possible neighbor VLC networks. The duration of the active period is referred to as active Superframe Duration (SD). At the same time, SD is subdivided into two parts, Contention Access Period (CAP) and Contention-Free Period (CFP). Under network operation, SD presents a fixed size. Hence, when CFP increases, CAP has to decrease. During CAP, the user nodes access the channel randomly using a back-off mechanism established on the standard, whilst during CFP they have assigned fixed-size time windows known as Guaranteed Time Slots (GTS). When a user node requires extra transmission capacity it needs to request additional slots to the coordinator. Furthermore, the assigned slots for each user are continuous, and the coordinator dynamically allocates GTS taking this requirement into account.

Moreover, the ratio between SD and BI (percentage of network activity) can be modified at the beginning of the network operation by setting both Super Order (SO) and Beacon Order (BO) parameters. The maximum value of BO and SO is 14, and SO cannot be higher than BO. SO increases the number of optical clocks per slot, taking as the initial value the base time *aBaseSlotDuration*. Since SD has a fixed number of slots (*aNumSuperframeSlots*), the difference between BO and SO will determine the aforementioned ratio. Equations (Equation 1) and (Equation 2) define the amount of optical clocks assigned to each of these periods:(1)$$SD=aBaseSlotDuration\times aNumSuperframeSlots\times 2^{SO}$$
(2)$$BI=aBaseSlotDuration\times aNumSuperframeSlots\times 2^{BO}$$

It is worth mentioning that all the processes in the MAC layer are based on the number of optical clocks. The relation between optical clocks and bit depends on the PHY configuration (modulation and encoding). Finally, if a specific ratio $\eta_{MAC}$ of channel utilization were intended, it is straightforward to demonstrate from the previous equations that the difference between BO and SO should follow Equation (Equation 3):(3)$$BO-SO=-\log_{2}\eta_{MAC}$$

### 3.2. Back-Off Algorithm

The access method during CAP is random. Thus, it requires a set of steps called back-off mechanisms to alleviate collision likelihood and decrease latency. This standard uses CSMA/CA, where the transmitter node needs to sense the channel before beginning its transmission to be sure that it is idle. The random access mechanism is based on the following steps. An initial random waiting time *backOffPeriod* is chosen randomly following a uniform distribution (Equation (Equation 4)):(4)$$backOffPeriod=aUnitBackoffPeriod\cdot U(2^{BE})$$

*aUnitBackoffPeriod* is the minimum back-off period and the Back-Off Exponent (BE) determines the width of uniform distribution U(·), which ranges from 1 to 2${}^{BE}$. After waiting *backOffPeriod* optical clocks, the user node senses the channel before transmitting as CSMA/CA suggests. In case of being unable to transmit, a failure is assumed, and a re-transmission is scheduled. If the channel is free, it transmits and waits for an ACK response to assess that the packet has been correctly delivered. When an unsuccessful transmission happens (ACK not received, or channel occupied, e.g.), a counter named Number of Backoffs (NB) is increased. NB is used for limiting the number of transmission attempts. Furthermore, the BE also increases (up to an upper boundary named *macMaxBE*), rising the maximum possible waiting time. The block diagram of this procedure is shown in Figure 2.

### 3.3. User Association

The most studied part of the MAC layer are back-off mechanisms (as the literature suggests) which limit the data-rate but improving the reliability of the network. Nonetheless, in networks with high node mobility, it is fundamental to evaluate the association process and back-off algorithms in the same simulation.

The association process begins when an un-associated node detects a beacon. After a random period, it sends an association request during CAP. The coordinator immediately sends an ACK frame to inform that the request has successfully arrived. At the same time, the coordinator sends a request to higher layers to determine source availability. When the ACK frame arrives at the node, the *macResponseWaitTime* counter is initialized. This time is longer than the ACK time because the resolution of the association request depends on several aspects such as scheduling algorithms or even the availability of network-related task time. When the request is resolved, a response arrives at the coordinator. It adds the new node’s address to its associated nodes’ table and sends the corresponding association response command to the node. The node stores its address and starts to use the network. The association process is described in Figure 3.

Besides, the coordinator expels users using a disassociation notification when it is not possible to support communication anymore or when there is a notification from an upper-layer. The standard does not define how the coordinator must physically communicate with upper layers or the specific frame formats.

## 4. Proposed Simulation Scheme

As mentioned in Section 2, most authors focused only on MAC modeling, using closed-form expressions to estimate path loss in the communications link. Nonetheless, VLC coordinators are usually located on the ceiling pointing downwards, limiting the likelihood of LOS situations between user nodes. However, depending on the scenario’s materials, reflections may be sufficiently powerful to be used during CAP to carry out CSMA/CA. In general terms, accurate impulse responses have not been generally considered during MAC-layer simulations, limiting the practical usefulness of those results. In this work, a combination of a MAC-layer simulation tool and a Monte Carlo Ray Tracing (MCRT) algorithm for realistic channel estimation is considered. This approach allows the evaluation of NLOS situations in which channel sensing could be feasible, as well as the possibility of considering real-world scenarios.

There is a significant number of programming platforms to carry out network-level simulations taking into account MAC operation and simplistic physical behavior. In this work, OMNET++ has been selected because it is based on Object-Oriented Programming (OOP), it is a C++ framework and is easily scalable. Since OMNET++ defines an event-driven simulation, it provides an easy interface to implement the MAC-layer operation based on callbacks.

The developed simulator is illustrated in Figure 4. It comprises of two main modules that reflect both the coordinator and nodes. Each of these modules is divided into two parts. The physical layer submodule evaluates the communication performance according to the link’s characteristics (which are accurately obtained using a MCRT algorithm), whilst the MAC control submodule applies the logical procedures of IEEE 802.15.7 and considers time synchronization.

A flow diagram of the simulation process is depicted in Figure 5. Each part of the algorithm is described in the following subsections.

In OMNET++ the modules present a hierarchical organization in which the main modules can be split into smaller ones in charge of simpler tasks. The small modules are associated with a C++ file and a NED file, whilst the complex modules only have a NED file. The C++ file contains the behavioral description of each node. On the other hand, the NED file describes its organization and allows simulation parametrization without modifying the C++ file. In this work, a complex module for the coordinator and another one for the user node have been developed. In these modules, it is possible to set up MAC and PHY parameters such as BO, SO, the optical clock, the node location, and the transmitted power, among others. To modify them, the new values can be added in the parameters section of each node’s NED file. Furthermore, the network is described in a separate NED file in which the considered nodes (coordinators and users) are included and connected.

### 4.1. Channel Simulation

MCRT allows the evaluation of the impact of surrounding objects in indoor scenarios such as walls, furniture, or even other users on the communications link. In this work, a Modified MCRT (MMCRT) has been used [8]. The algorithm randomly generates a set of rays from a light source following the radiation pattern as the probability density function. These rays travel through the room impinging on the scenario’s surfaces. After each rebound, a forced contribution is calculated and the rest of the energy is scattered taking into account the reflection pattern. The developed channel simulation tool receives a triangularized mesh file of the scenario (.obj file), and a table relating each material with its reflection pattern.

The simulator has been divided into three main parts: Pre-processing, MMCRT calculation, and post-processing. During the pre-processing stage, an OCTREE (octal tree) structure is inflated using the triangularized mesh file that describes the scenario’s geometry. The use of this structure enhances impact calculation performance, since it implements a logarithmic-complexity search algorithm. The MMCRT stage carries out the calculation of the ray contributions on each target point, using Equations (Equation 5) and (Equation 6) for the LOS and NLOS contributions:(5)$$h{\left(t,{\hat{n}}_{imp}\right)}_{LOS}=P_{tx}\left(\theta_{LOS},\varphi_{LOS}\right)\frac{1}{d_{LOS}^{2}}\delta \left(t-d_{LOS}/c_{0}\right)\cdot {\hat{n}}_{LOS}.$$

$\theta_{LOS}$ and $\varphi_{LOS}$ are the ray emission angles from the transmitter’s reference, whilst ${\hat{n}}_{LOS}$ is the direction vector of the LOS ray. $c_{0}$ is the speed of light, and $d_{LOS}$ is the LOS link range:(6)$$h{\left(t,{\hat{n}}_{imp}\right)}_{NLOS}=\frac{1}{N}\sum_{i=1}^{M}R_{K(i)}\left(\theta_{i,K(i)},\varphi_{i,K(i)}\right)\delta \left(t-\sum_{j}d_{j}/c_{0}\right)\cdot {\hat{n}}_{i,K(i)}\prod_{j=1}^{K(i)}\frac{\rho_{j}}{d_{j}^{2}}\frac{1}{L}.$$

The *M* arriving rays that conform to the NLOS contribution of the Channel Impulse Response (CIR) suffer from a different number of rebounds. Each random ray is weighed by a number of random directions generated on each rebound. *N* rays are generated at the transmitter and *L* after each impact. Since an emphasized random sampling scheme is used in MMCRT, the radiation pattern only has to be taken into account on the forced contributions. $R_{K(i)}\left(\theta_{i,K(i)},\varphi_{i,K(i)}\right)$ is the reflection pattern of the *i*-th ray’s last rebound K(i) at the corresponding angles. Finally, $\rho_{j}$ is the reflectivity of the *j*-th surface, and ${\hat{n}}_{i,K(i)}$ is the direction of the last rebound (surface-to-receiver). Nevertheless, the obtained response does not take into account the receiver optics since it only stores the impact angle, power, and flight time. Hence, the result is a tensor that defines an angle-dependent impulse response $h(t,{\hat{n}}_{imp})$, where ${\hat{n}}_{imp}$ is the impact direction vector. Finally, during the post-processing stage the tensor-like impulse response is projected into a time-power signal using the description of the receiver optics and its attitude (Equation (Equation 7)):(7)$$h\left(t\right)=A_{pd}G_{lens}\int h\left(t,{\hat{n}}_{imp}\right)\cdot {\hat{n}}_{rx}\;\Theta \left({\hat{n}}_{imp}\cdot {\hat{n}}_{rx}-\cos\left(FOV/2\right)\right)\;d{\hat{n}}_{imp}.$$

$A_{pd}$ is the receiver’s active area, $G_{lens}$ is the lens gain, and Θ is Heaviside’s theta. In essence, the effective area of each impact is calculated taking into account the receiver’s attitude ${\hat{n}}_{rx}$, and those rays outside the receiver’s FOV are neglected. The proposed three-stage algorithm provides improved flexibility, since time-variant receiver attitude (due to head movement, e.g.,) can be considered without executing the whole time-consuming algorithm again.

Among the channel parameters that the impulse response provides, DC channel gain H(0) and bandwidth *B* are the most relevant, since they define the quality of the assumed linear system.

To provide an estimation of $H_{0}$ it is necessary to aggregate the contributions from all the received rays. Moreover, using the link loss it is possible to determine the Signal-to-Noise Ratio (SNR) at the receiver considering both shot and thermal noise (Equation (Equation 8)):(8)$$SNR=\frac{{\left(P_{tx}H\left(0\right)R_{\lambda }\right)}^{2}}{\sigma_{th}^{2}+2q\left(P_{tx}H\left(0\right)R_{\lambda }+i_{d}+i_{b}\right)B_{N}}.$$

$R_{\lambda }$ is the responsivity of the receiver, $\sigma_{th}^{2}$ is the Johnson noise, *q* is the electron charge, $i_{d}$ is the receiver’s dark current, $i_{b}$ is the background-induced current, and $B_{N}$ is the noise bandwidth of the receiver. Traditional OWC systems work using a single wavelength (typically IR). Nonetheless, VLC systems use broadband optical emitters and the whole range of wavelengths should be used to calculate the signal power (numerator of the SNR). Nonetheless, an approximation can be carried out using the average wavelength [27].

The more scattered the CIR contributions are, the lower the channel bandwidth. The Root Mean Square (RMS) delay spread $\tau_{rms}$ describes the channel’s time dispersion (Equation (Equation 9), and can be used to get an approximation of the channel bandwidth *B* (Equation (Equation 10)).
(9)$$\begin{array}{cc} \tau_{rms} & ={\left(M_{2}-M_{1}^{2}\right)}^{1/2}\\ M_{i} & =\frac{1}{H(0)}\int_{0}^{\infty }\tau^{i}h\left(\tau \right)\;d\tau \end{array}$$
(10)$$B\approx \frac{1}{5\tau_{rms}}$$

Most MAC simulations do not take channel bandwidth into consideration. However, if the channel cannot support the minimum bandwidth requirements, the communication link would not be established. Situations in which a receiver does not have LOS with the access point may suffer from a reduced bandwidth performance. In this work this is taken into account to provide more realistic simulations.

Commonly, these simulations use a huge number of rays to obtain high precision. Nonetheless, for MAC evaluation a faster calculation may provide sufficient information to carry out the simulation, since a rougher estimation of the parameters is enough to assert link quality. In this work, the channel estimation has been carried out using a small set of 500 rays. Despite the reduced number of rays, the accuracy of the estimation is not significantly impaired in terms of channel gain and bandwidth thanks to the use of a MMCRT algorithm that forces direct contributions after each rebound. This algorithm has been validated compared to Barry’s recursive method, which is considered as the gold standard in OWC [28].

Although the simulator provides point-to-multipoint impulse responses without a significant time penalty (using a simultaneous calculation in a map of points), the number of simulations that must be performed is equal to the number of elements in the network. Each node (coordinator and users) inside the simulation environment needs to check the channel with all the other nodes. The sampling rate of the CIR between each link evaluation would depend on the nature of the scenario.

### 4.2. Physical Layer Sub-Module

The PHY sub-module evaluates the communication link using the results of the channel simulation process using Equations (Equation 8) and (Equation 10). First of all, the received power is compared to the endpoint sensitivity. If the power is lower than this threshold, this module marks the packet as lost. On the other hand, it is checked by the collision detection block of the submodule, and it is determined if the packet has been correctly received. Using the SNR, the Bit Error Rate (BER) can be easily calculated depending on the configuration of the VLC PHY transmission mode. For a given packet length $N_{packet}$, the number of errors $N_{e}$ in the frame follows a Binomial distribution (Equation (Equation 11)):(11)$$N_{e}∼B\left(N_{packet},BER\right).$$

Depending on the error-correction scheme of the transmission mode, the packet would be marked as correctly received or not, appropriately informing the upper layer.

The collision detection block checks whether the transmission was completed without interference from other users or not. This block retains the frame during its transmission period. If another packet arrives during this period, a collision occurs, and both packets are destroyed. Then, a new waiting time is calculated considering the end time of the latest transmission. This period depends on the frame size, the optical clock duration, and the used modulation scheme. If the waiting time expires and there was not any collision, the frame is delivered to the MAC layer sub-module to be handled. This sub-module additionally updates the node positioning if necessary. The described sub-module structure and process is illustrated in Figure 6.

#### Mac Layer Sub-Module

The MAC layer sub-module performs the logical decisions that each node makes during the protocol execution. The sub-modules of both the coordinator and user nodes behave differently. One of the most relevant discrepancies is the synchronization block. The coordinator’s sync block transmits beacon frames continuously to provide a time reference to the users, whilst the user’s sync block receives these beacon frames and operates accordingly. When it is not possible to receive a beacon (due to collisions or poor channel performance) this sub-module stays idle.

When a frame arrives from the physical layer, the MAC layer checks the frame type. This stage categorizes the frame by its function (Data, ACK, Command, or Beacon) and relies it to the correct block to be processed. Command frames are dispatched to the MAC control block, the actions performed by this block depend on the message itself, and are related to different mechanisms of the standard that were described in Section 3. On the other hand, data frames need to be handled simultaneously by the temporal filter block and the ID control block. The temporal Filter block verifies that user packets are in their correct superframe window. If they arrive when the packet is not supposed to, they are discarded. The ID control block verifies the ID of the sender and recipient. The coordinator’s block checks if the sender is allowed to use this access point. In case that the recipient is associated with it, the frame is sent using the VLC channel, otherwise it is passed to upper layers. At this point, the statistics generated during the packet transmission are collected. Finally, the output control block handles the data transmission, and it can operate in CAP or CFP mode according to the user’s resources. This block executes the random back-off mechanism and queues packets. Figure 7 illustrates the organization diagram of this sub-module.

In the simulator, it is possible to add more communication layers. Nevertheless, in this work, a network encompassing just the MAC and PHY layers has been evaluated. The rest of the layers were simplified, only generating data to be transmitted and collecting metrics for the evaluation.

## 5. Methodology

The main objective of this work is to demonstrate that the proposed simulation scheme, which integrates both MAC layer operation and a realistic optical physical layer estimation, provides more insight about the performance of IEEE 802.15.7 VLC networks than traditional MAC-only simulations. In order to validate this hypothesis, different network topologies and physical scenarios have been taken into account. The following subsections describe the experimental setup, the procedures that have been carried out, and how the obtained data have been analyzed.

### 5.1. Experimental Setup

The experimental setup is comprised of different network topologies. Each network deployment is used for evaluating different situations that are likely to occur in OWC, such as the effect of multipath reflections in the CSMA/CA scheme, or the hidden node problem associated to the receivers’ limited FOV.

To validate the importance of considering a realistic channel model on the network evaluation, three different cases have been considered. In these scenarios, there is a VLC network composed of 1 coordinator node and 4 user nodes inside a 10 × 10 × 3 m${}^{3}$ room. All the nodes have a receiver with a 1 cm${}^{2}$ photodiode and FOV of 60${}^{\circ }$ Full Width Half Maximum (FWHM). The coordinator is at the room’s geometrical center (in the XY plane) and at the ceiling pointing downwards. The user nodes are set at 1 m height, and their transmitter and receiver are both pointing upwards. These nodes are located at the same radial distance regarding the coordinator’s XY center. However, this baseline condition is slightly modified on each case under study. In the first case, all the nodes are concentrated in a specific area of the room. This scenario forces a situation in which the user nodes may have sufficient reflected power from the ceiling to properly operate using the CSMA/CA mechanism. In the second case, they are uniformly distributed on a circumference, forming a square. In this case, depending on the radii the user nodes will be aware of other nodes. Finally, in the last case three nodes are closely located whilst the other one is at the opposite direction. This situation is proposed in order to see the impact of a single hidden node on the network’s performance. Figure 8 depicts the node distribution of each case and a 3D representation of each one of them.

As it was aforementioned, at short distances the reflection allows the correct execution of the MAC mechanisms. Nonetheless, as the nodes increase their distance, the path loss may become too high, lowering the received signal below the endpoint’s sensitivity. Figure 9 shows the channel gain for the communications link between two nodes in these scenarios, using the parameters of Table 1.

The number of rays used for the MMCRT-based simulation was chosen attending to empirical criteria. One of the strengths of the used ray tracing scheme is that a contribution to the impulse response is forced after each rebound, taking advantage of the underlying physical behavior of reflections (energy is scattered and some energy is directed towards the receiver). Hence, it seemed straightforward that obtaining a relatively accurate estimation of both channel gain and bandwidth could be feasible with a reduced number of rays. To check this assumption, some simulations using the same layout as in Figure 9 were performed. As it can be observed in Figure 10, neither the DC channel gain nor the bandwidth changed substantially compared to simulations using a higher number of rays (1000 and 5000). The estimated channel gain using 200 or 500 rays is practically indistinguishable from longer simulations. Nonetheless, bandwidth estimation is significantly different for the 200 rays case, whilst acceptable for the 500 rays calculation. Therefore, for the scenarios under consideration, carrying out the channel simulation using 500 rays is a trade-off between calculation speed and accuracy. In addition, despite the slight variation of the bandwidth estimation with respect to more accurate simulations, the presented error is sufficiently small for moderate horizontal node separations, which are the ones at which the CSMA/CA mechanism can physically operate taking into account the sensitivity of the endpoints.

Moreover, in order to properly characterize the network, both saturated and unsaturated traffic conditions have been tested. As it was used in other works [15,17], 70% of the maximum traffic capacity of the network was established as a threshold to determine the saturation condition. For the unsaturated network condition, the frame arrival rate (or frame generation rate) of the nodes corresponded to a exponential distribution with a mean time of 953.6 μs. These frames presented a payload of 2000 bits. Considering the ACK time, header size, and the amount of users, this configuration demands approximately a 20.54% of the maximum manageable traffic. On the other hand, the saturated configuration presented a mean time of 100 μs, resulting in beyond-capacity necessities (195% of the maximum traffic). This a priori huge value was selected to keep a very demanding situation even when some of the nodes were unable to be associated.

Some parameters of IEEE 802.15.7, concretely the association parameters, were chosen according to other works [29,30] since there is no guide or reference for them. The association request was sent randomly with a 38 slots window. This request must be completed before 300 ms, or the node assumes that it failed and repeats the operation.

### 5.2. Description of the Validation Procedure

As commented above, three different scenarios targeting different effects were defined. Each one of these scenarios was evaluated for a range of radii (geometrical parameter of the scenarios). Furthermore, two different ceiling materials, presenting different reflectivity coefficients, and two frame arrival rates (saturated and unsaturated traffic) were also analyzed. The simulator was run 15 times in order to obtain statistically significant results, and the extracted metrics were compared to a MAC-only (no path loss) simulation.

Considering all the different experimental setups 2160 simulations were carried out. Each one of these simulations recreated the behavior of an IEEE 802.15.7 VLC network during 100 s, generating between 4·10${}^{5}$ and 4·10${}^{6}$ messages per simulation (depending on the simulation parameters). This huge amount of information is enough to obtain consistent statistical conclusions. The simulations were carried out without parallelization in a workstation with an Intel Core i7 (1.99 GHz) and 16 GB of DDR3 RAM. For the non-saturated traffic scenarios each simulation was executed in approximately 50 s, whilst for the saturated traffic ones 170 s were spent per execution.

At the beginning of the execution, all user nodes were marked as un-associated and all the internal variables of the simulation modules were set to their default state. Once the simulation started, the nodes began to perform association requests until they were properly notified by the coordinator node. After a successful association, each node started to generate 2000 bit packets (payload size) at the rate defined by the scenario under evaluation. Each packet was handled as commented in Section 4, using a frame buffer of 50 packets.

Each event that occurred during the 100-s simulation was logged into a file, describing the emitter, the receiver, the received power (if applicable), the event type, and a timestamp. All this information was used by the data analysis stage to extract the metrics of this work, which are commented in the following subsection.

### 5.3. Data Analysis

The metrics obtained from the simulation were collected as follows.

**Throughput**. It is the aggregated traffic that arrives to the coordinator, and calculated as the amount of payload bits obtained from data frames divided by the simulation time;**Queued Packet Delivery Probability (QPDP)**. This is a measure of the network availability for those packets that have not overflowed the node’s frame buffer. It is calculated as the total number of correctly delivered packets divided by the amount of queued packets;**End-to-end Packet Delivery Probability (EPDP)**. This measure is analogous to the latter, but it also takes into account the discarded packets due to excess of traffic on each user node. Therefore, EPDP should be strictly lower than QPDP since it is calculated as the total correctly received frames divided by the amount of generated frames;**Node active time**. This metric indicates the amount of time that a node has been correctly associated to the network. It is calculated as the difference between the timestamps related to periods in which the user node is correctly associated to the network;.**Delivery time**. It is the time that the communication system spends to successfully deliver a frame. This measure is calculated as the difference between the timestamps of the first delivery attempt of a message and its correct reception.

These metrics would depend on the simulation parameters such as the deployment radius and the amount of traffic as expected. Furthermore, the impact of the ceiling material was assessed through a parametric hypothesis test. This hypothesis has not been previously analyzed (up to the authors’ knowledge), and would serve to validate the proposed comprehensive simulation scheme. Concretely, Welch’s T-Test was performed to compare the average values of both scenarios according to Equation (Equation 12). The null hypothesis to be rejected in these cases was that the material had no impact. Hence, a bilateral test was carried out:(12)$$T=N\frac{{\bar{X}}_{w}-{\bar{X}}_{p}}{\sqrt{\sigma_{w}^{2}+\sigma_{p}^{2}}}.$$

*N* is the number of simulations, ${\bar{X}}_{w}$ and ${\bar{X}}_{p}$ are the average of metric *X* (throughput, probabilities, etc.) for wooden and plaster ceilings correspondingly, and $\sigma^{2}$ is the quasi-variance. Moreover, no equality of variances is assumed. Hence, the degrees of freedom (dof) of the null hypothesis model (Student-T distribution) were estimated using Welch–-Satterthwaite’s approximation (Equation (Equation 13)):(13)$$dof\approx \left(N-1\right)\frac{{\left(\sigma_{w}^{2}+\sigma_{p}^{2}\right)}^{2}}{\sigma_{w}^{4}+\sigma_{p}^{4}}$$

Both equations have been simplified considering that both populations are comprised the same number of elements (*N*). The contrast statistic *T* was ultimately used to obtain an estimation of the *p*-value (or probability of failing after rejecting the null hypothesis).

## 6. Results

The results section has been subdivided considering each one of the metrics of interest defined in Section 5. The impact of the evaluated parameters on each metric is presented and in-depth discussed. Moreover, the parametric hypothesis test results on the impact of the ceiling material are presented in the final subsection.

### 6.1. Throughput

The network throughput is the result of aggregating all the correctly delivered traffic at the coordinator node. It is expected that situations in which the user nodes are capable of properly sensing the channel, this metric is maximized. On the other hand, as the CSMA/CA performance gets impaired by an excessive link range between nodes (or even between coordinator and node), it is expected that the throughput decreases. Figure 11 depicts the obtained throughput for the two evaluated traffic conditions.

It can be observed that Scenario 1 presents a performance close to the MAC-only simulation for a radius below 2.5 m. The reduced link range between coordinator and user nodes, and the advantageous location of the nodes in terms of CSMA/CA operation contribute to this result. Scenario 2 presents a similar behavior, but the distance at which throughput starts to diminish in respect to the MAC-only simulation is smaller. This occurs because the nodes placed on a regular mesh lose their mutual visibility before the coordinator-node link is affected by the geometry. In respect to Scenario 3, it is observed that it has the worst performance due to the presence of a hidden node. The commented trends apply to both traffic conditions, but the difference between Scenario 3 and Scenario 1 is sharper for saturated traffic than for reduced traffic necessities. This higher impact on performance is due to the effect of delivery probability, which is presented in the following subsection.

### 6.2. Delivery Probability

In this subsection, two different metrics are presented. On the one hand, QPDP takes into account only those messages that were accepted in each node’s buffer. On the other hand, EPDP also considers the discarded messages due to traffic overflow. As commented, EPDP must be strictly lower than QPDP. The relation between both metrics provides an estimation of the amount of lost traffic because of excessive buffering time. Figure 12 depicts QPDP, whilst Figure 13 illustrates EPDP results.

It can be observed that once a message is queued, the probability of correctly delivering it is above 82% in all the scenarios. In the case of unsaturated traffic, the impact of distance on this metric is quite reduced, due to the number retransmissions. However, as commented below, both delivery time and EPDP increased dramatically with distance. An increment on the delivery time would imply more channel resources per message, which may tend to a buffer overflow. It is also observed in Figure 12 that under saturated traffic conditions the QPDP is significantly higher for Scenario 1 than for the MAC-only situation. This exotic situation is derived from the increased disconnection probability at those distances. During some periods, due to the disconnection, the remaining nodes have more relaxed CAP restrictions.

As commented above, EPDP is smaller than QPDP. For unsaturated traffic, the trend is similar for all the three scenarios, presenting an EPDP around 0.9 at 3 m. The number of required retransmissions starts to be significant at this distance and some traffic is lost at each node due to overflow. On the other hand, traffic-saturated scenarios presented EPDP values below the MAC-only reference simulation in all the cases. As commented during the throughput subsection, there are distances at which collisions may occur with a higher probability at each scenario. Moreover, the shape of throughput and EPDP curves are closely related as expected.

### 6.3. Node Active Time and Delivery Time

Node active time measures the time spent by a node in the VLC network, whilst delivery time indicates the time difference between a correct reception and first try. Node active time is presented normalized in respect to the simulation time. Hence, it corresponds to the network availability from the node’s viewpoint. It must be taken into account that frame generation does not stop during disconnection periods. Therefore, reduced node active times may also reduce the EPDP due to packet losses. Figure 14 and Figure 15 present the normalized node active time and delivery time respectively.

Comparing saturated and unsaturated traffic conditions, it is observed that demanding traffic requirements affected node active time, slightly diminishing it. A larger collision probability due to extensive channel occupation led to an increased disconnection probability.

Regarding delivery time, it can be observed in Figure 15 that Scenario 1 conserves a good performance for all the link ranges, thanks to a proper carrier sensing. However, Scenario 2 loses performance beyond 2.5 m approximately since visibility between nodes is lost. This trend is broken in Scenario 3, which only has a single hidden node. In this case delivery time is conserved regardless of link distance. The minimum theoretical time a packet spends in the queue (considering the 50-slots buffer) is approximately 14 ms in this work. Depending on the application, the resulting overall latency (delivery time plus queue time) may be unacceptable, and GTS should be used instead.

### 6.4. Impact of Reflectivity

All previous results have considered two different ceiling materials. In this subsection, the impact of the distinct reflectivity values of wood and plaster is statistically assessed. These two materials were selected because they can be easily found in realistic scenarios. Nonetheless, this methodology is applicable to any material regardless its reflection pattern. Table 2 shows the *p*-values associated to each one of the scenarios.

The presented table provides the numerical results of carrying out Welch’s *t*-test. Only sufficiently small *p*-values are shown since they indicate that considering different materials in the simulation environment produces statistically distinct results. It can be observed that the outcomes of the joint MAC/PHY simulation procedure proposed in this work suggest that there are significant differences between the two considered materials in some cases. It can be also observed that this differences were very slight for Scenario 1. This occurred because the essayed reflectivities, although they are not very similar, allowed all the nodes to conserve their mutual visibility regardless of the ceiling’s material taking into account the scenario’s geometry. Nevertheless, Scenario 2, which is a priori the most radius-sensitive configuration, presented a significant amount of cases in which it is statistically proven that there is a difference between plaster and wooden ceilings. Finally, the analysis showed that Scenario 3 had more differences than Scenario 1, but less than Scenario 2 since there is only a single hidden node. These results demonstrate that the network performance metrics are susceptible to be affected by the physical properties of the scenario.

## 7. Conclusions

In this work, a comprehensive MAC/PHY simulation environment for the IEEE 802.15.7 standard based on OMNET++ and an ad hoc MMCRT algorithm has been presented. Traditional network simulation software considers only the MAC operation and in some cases a simplistic physical channel approximation, which may lead to unrealistic outcomes. The main hypothesis that underlies this work is that considering the physical channel is capital for obtaining realistic network metrics. In order to prove this, three scenarios presenting three different network situations (no hidden nodes, a single hidden node, and no visibility between users) were essayed under saturated and not saturated traffic conditions. Furthermore, these network scenarios were simulated using two different physical scenarios, which showed unequal reflection coefficients on their ceilings. Several metrics (throughtput, node active time, delivery time, and packet delivery probabilities) were obtained after logging all the events of the 15 simulations that were executed for each case, and they were statistically compared using Welch’s *t*-test.

The developed network simulation environment comprised two main modules. The channel simulation module was based on the aforementioned MMCRT algorithm, and was used to obtain the channel impulse response of each link and therefore both channel gain (or path loss) and bandwidth. This information was used by the network modules (developed for OMNET++), which consisted of a PHY-layer submodule and a MAC-layer one. The PHY submodule detected collisions, simulated errors on the received frames and delivered messages to the MAC submodule. On the other hand, MAC operation was simulated following IEEE 802.15.7 recommendations, including random back-off mechanisms, superframe synchronization, and queuing.

The obtained simulation results provided enough at-sight evidence about the differences between scenarios. Scenario 1 (no hidden nodes) was the closest to a MAC-only simulation since the nodes presented good CSMA/CA operation. However, as the network radii increased, the loss of coordinator-to-node link quality impaired performance on all the studied metrics. Scenario 2 showed similar metrics to Scenario 1 for small radii, but the metrics dramatically decreased above a threshold distance associated to a loss of visibility between nodes. On the other hand, Scenario 3 showed the worst behavior in general terms, suggesting that hidden nodes pose a likely harmful situation in VLC networks. From the traffic stress test carried out during the simulations, it can be concluded that a CAP-based network access could not be enough to allocate demanding applications in VLC. It was observed that although node active time was not affected by this parameter as expected, traffic accumulation led to massive packet losses, impairing QPDP and EPDP.

In order to prove that the physical scenario had a statistically significant impact on the network performance a hypothesis test on difference of means was carried out for all the obtained results. The results of the Welch’s *t*-test suggested that the physical environment affected most situations. Scenario 1 simulations, since they are proximal to a MAC-only case, presented no difference. Nonetheless, Scenarios 2 and 3, which are more intricate and geometry-dependent showed dependence on the scenario’s material.

This work has provided enough scientific evidence about the necessity of considering accurate channel estimation in VLC network simulations. Furthermore, the developed simulation environment is easily scalable due to its modular nature and flexibility thanks to the use of fast MMCRT algorithms that will enable an analysis of mobility scenarios in further research.

## Figures and Tables

**Figure 1 sensors-20-06014-f001:**
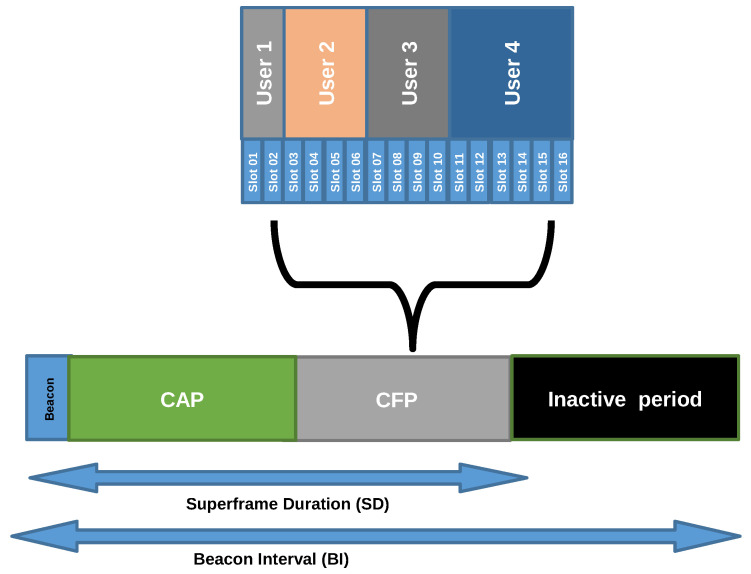
The superframe structure of IEEE 802.15.7 standard.

**Figure 2 sensors-20-06014-f002:**
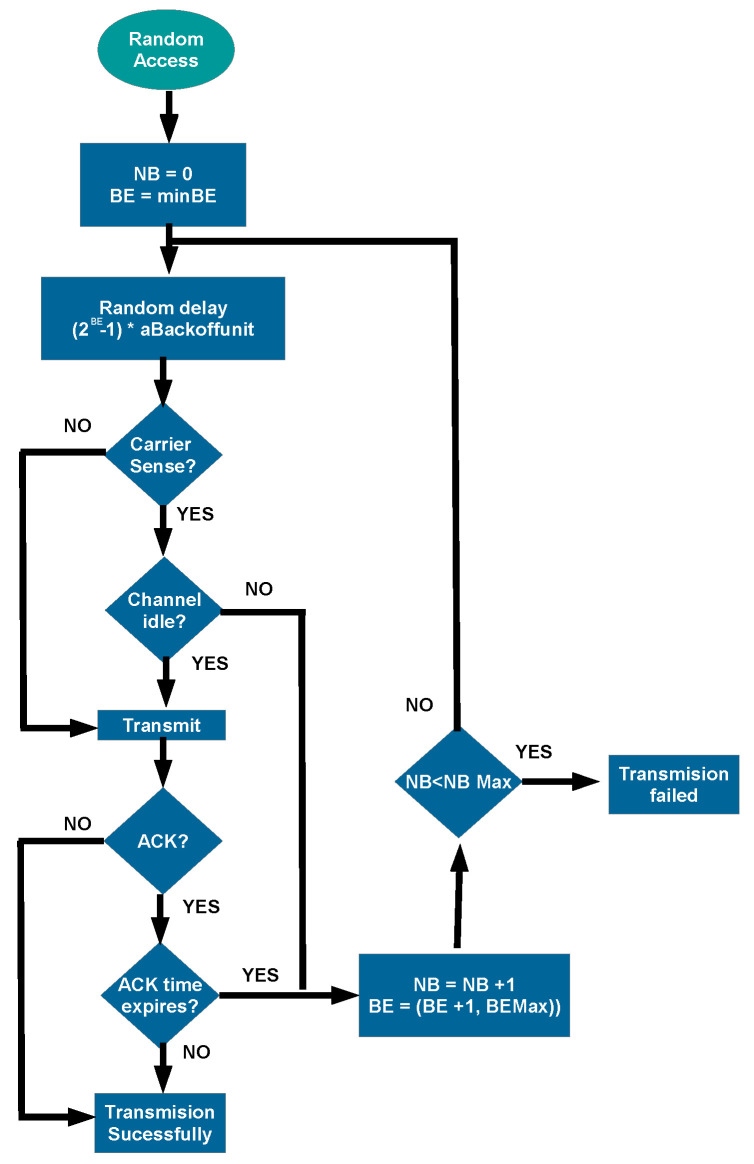
The back-off algorithm during the Contention Access Period (CAP) of standard IEEE 802.15.7.

**Figure 3 sensors-20-06014-f003:**
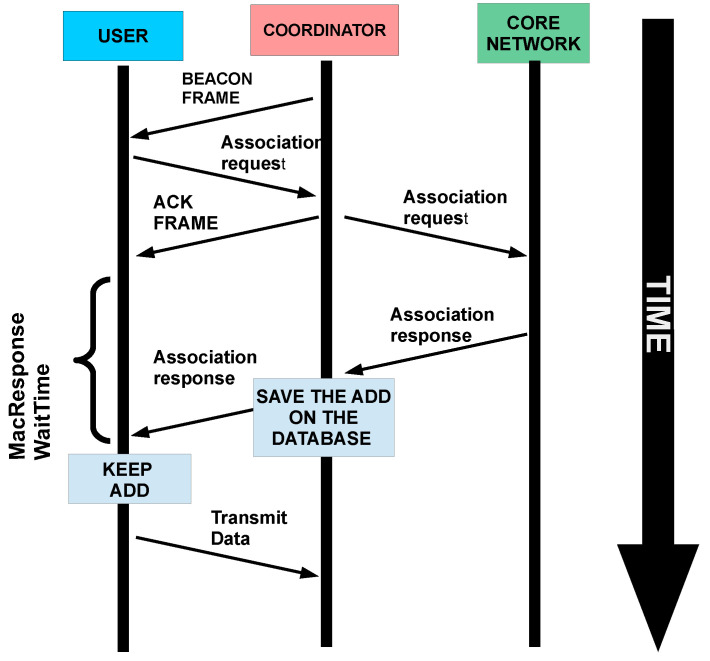
Association process of a new node in a Visible Light Communications (VLC) network.

**Figure 4 sensors-20-06014-f004:**
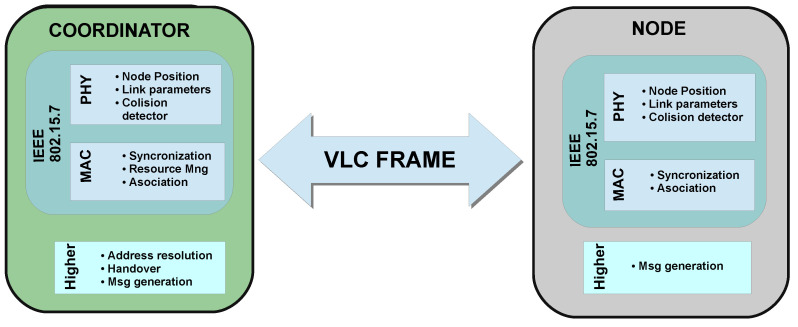
Structure of the developed simulation platform.

**Figure 5 sensors-20-06014-f005:**
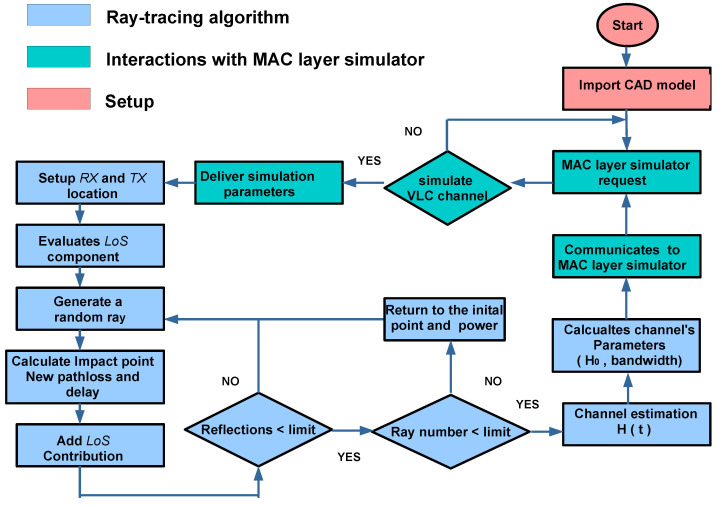
Flow diagram of the Monte Carlo Ray Tracing algorithm for physical layer simulation.

**Figure 6 sensors-20-06014-f006:**
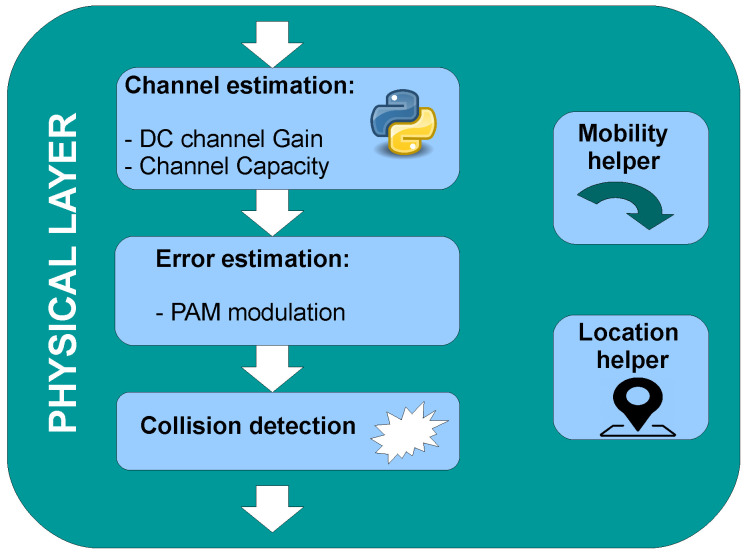
Block diagram of the physical layer sub-module.

**Figure 7 sensors-20-06014-f007:**
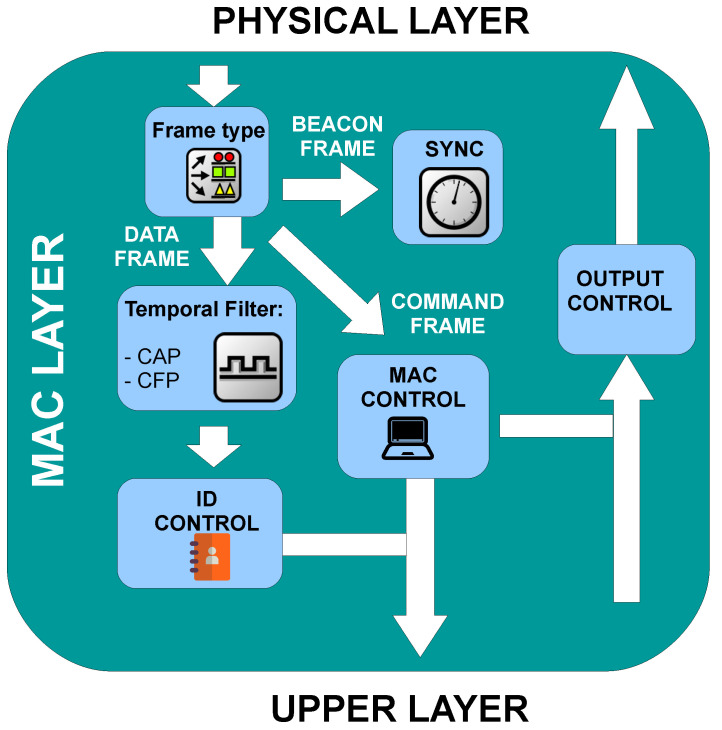
Block diagram of the MAC layer sub-module.

**Figure 8 sensors-20-06014-f008:**
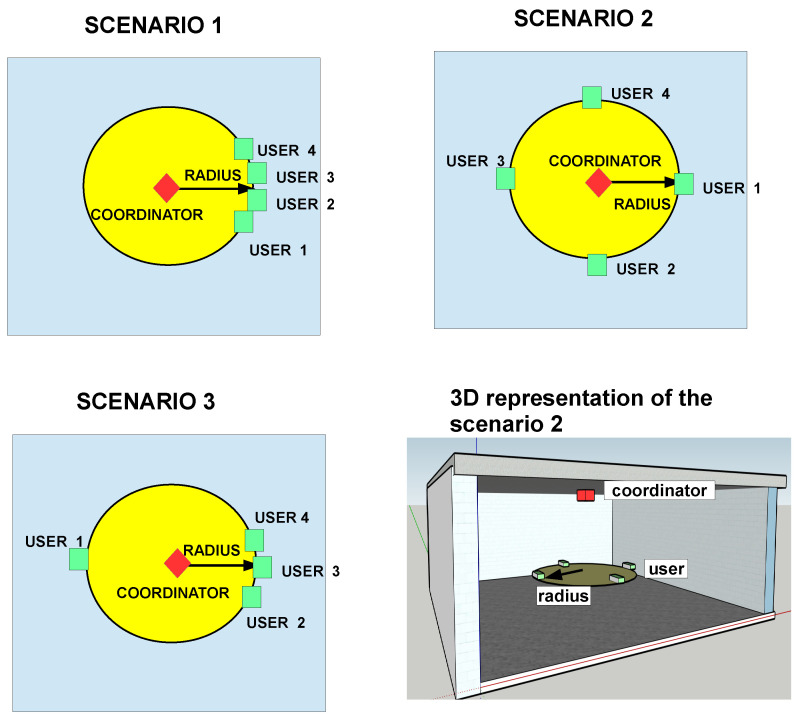
Graphical representation of the scenarios setup.

**Figure 9 sensors-20-06014-f009:**
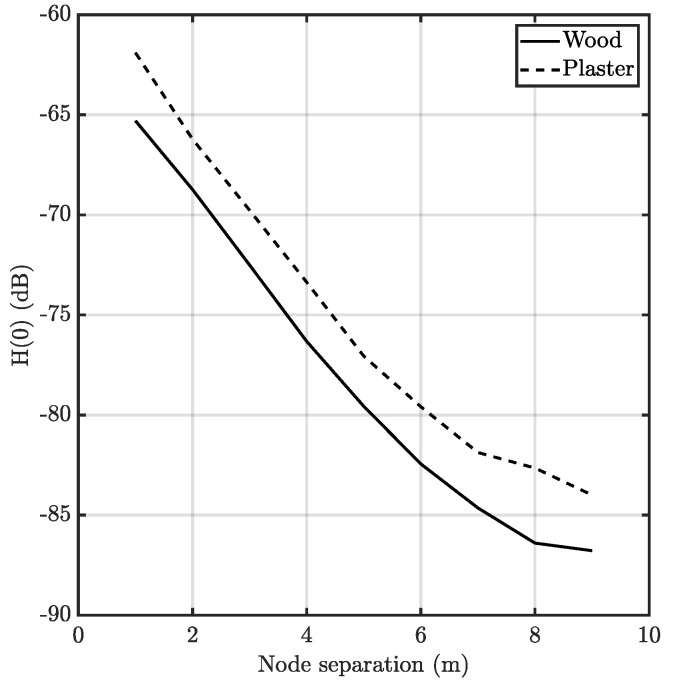
DC channel gain of the Non-LOS (NLOS) link between nodes at different horizontal distances.

**Figure 10 sensors-20-06014-f010:**
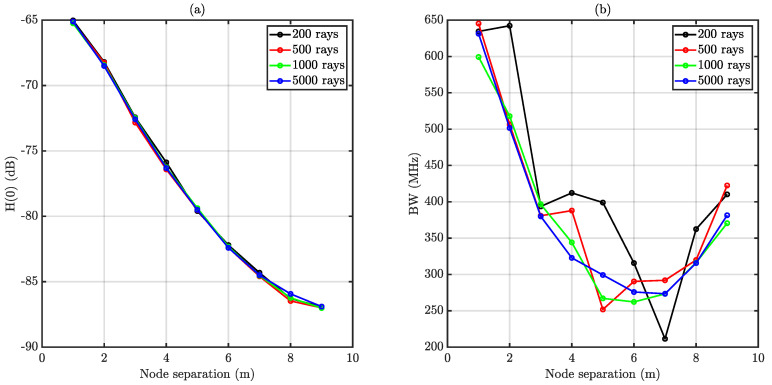
Impact of the amount of rays on the outcomes of the Modified Monte Carlo Ray tracing (MMCRT) simulation. (**a**) illustrates DC channel gain whilst (**b**) corresponds to bandwidth.

**Figure 11 sensors-20-06014-f011:**
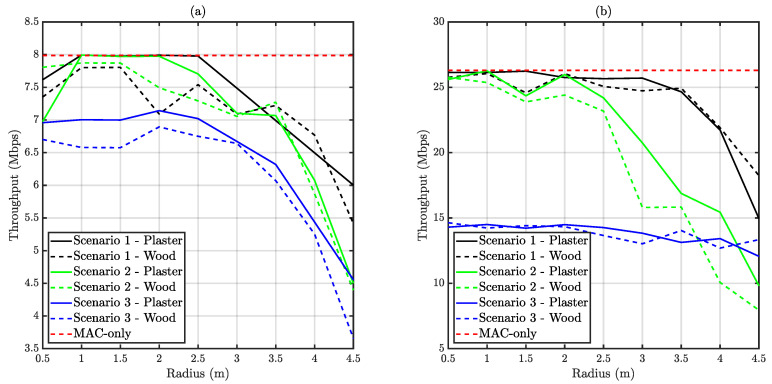
Obtained average throughput versus distance. (**a**) Depicts unsaturated traffic whilst (**b**) corresponds to saturated traffic.

**Figure 12 sensors-20-06014-f012:**
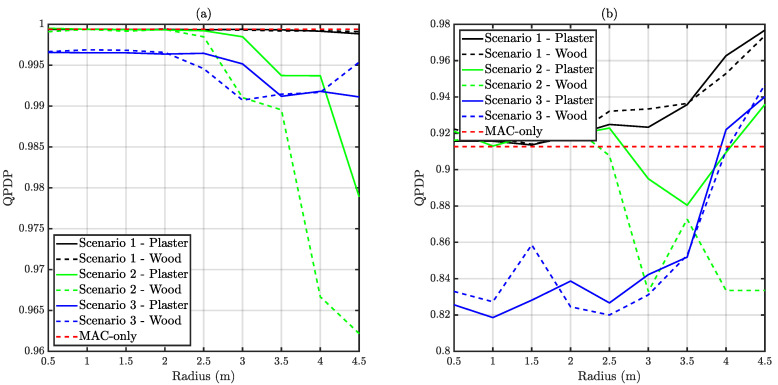
Obtained average Queued Packet Delivery Probability (QPDP) versus distance. (**a**) Depicts unsaturated traffic whilst (**b**) corresponds to saturated traffic.

**Figure 13 sensors-20-06014-f013:**
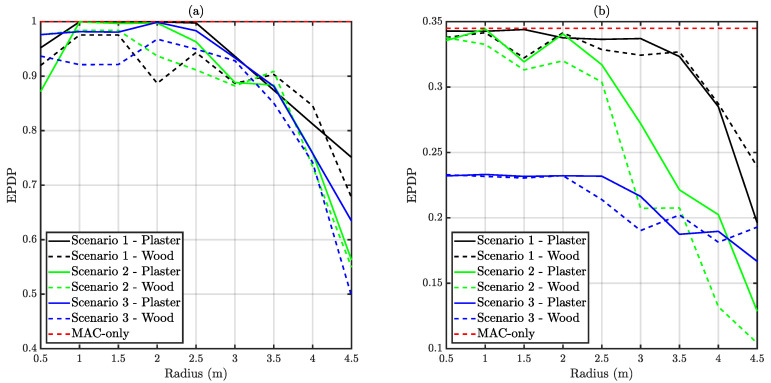
Obtained average End-to-end Packet Delivery Probability (EPDP) versus distance. (**a**) Depicts unsaturated traffic whilst (**b**) corresponds to saturated traffic.

**Figure 14 sensors-20-06014-f014:**
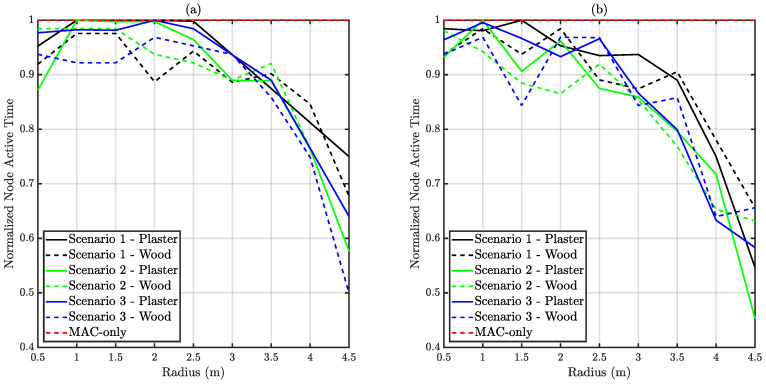
Obtained average normalized node active time versus distance. (**a**) Depicts unsaturated traffic whilst (**b**) corresponds to saturated traffic.

**Figure 15 sensors-20-06014-f015:**
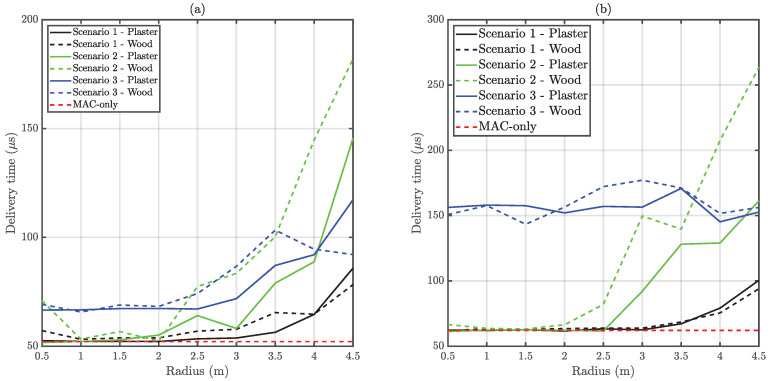
Obtained average delivery time versus distance. (**a**) Depicts unsaturated traffic whilst (**b**) corresponds to saturated traffic.

**Table 1 sensors-20-06014-t001:** Summary of simulation parameters. This table includes information about the scenario and both PHY and MAC layers.

Parameter	Value	Parameter	Value	Parameter	Value
Room size	10 m × 10 m × 3 m		6	Number Backoff	5
Room’s material	plaster/wood	SO	6	Number of slots	16
Reflection coefficient	0.75/0.4	Frame payload	2000 bits	Modulation	OOK
Number of users	4	Frame header	270 bits	Code	10B8B
Number of coordinators	1	ACK length	50 bits	LED Lambertian order	1
Radii	0.5 to 4.5 m	Optical Clock	60 MHz	aBaseSlotDuration	60
Number of rays in MMCRT simulation	500	Random time window for an association request	38 slots	Simulation time	100 s
Reflections per ray	3	MAC response wait time	300 ms	Number of simulations	15
Receiver area	1 cm^2^	Backoff unit	200	Coordinator Tx power	200 mW
Receiver FOV (FWHM)	60${}^{\circ }$	maxMaxBE	3	Node Tx power	50 mW

**Table 2 sensors-20-06014-t002:** Probability of failing after rejecting the null hypothesis for each one of the evaluated scenarios and metrics. Only the *p*-values smaller than a significance level of 0.1 are shown (the radii are between parentheses). * equals lower than 0.0001.

Metric	Scenario 1	Scenario 2	Scenario 3
Throughput		0.0533 (2 m) 0.009 (3 m) 0.0394 (4 m)	0.028 (2.5 m)
QPDP		0.019 (3 m) 0.024 (4 m) 0.003 (4.5 m)	0.1 (1 m) 0.074 (1.5 m)
EPDP		0.054 (2 m) 0.009 (3 m) 0.039 (4 m)	0.0 * (2.5 m) 0.003 (3 m)
Node Active Time			0.088 (1.5 m)
Delivery Time	0.011 (1 m)	0.077 (0.5 m) 0.0 * (2.5 m) 0.0 * (3 m) 0.0 * (4 m) 0.0 * (4.5 m)	0.0 * (2.5 m)

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
