# Peer review of "MAC/PHY Comprehensive Visible Light Communication Networks Simulation"

_sensors, 2020, doi:10.3390/s20216014_

Round 1
Reviewer 1 Report
The article is well written.
I feel there are some minor proofreading issues.
I would like to make a suggestion to make the like of the figure a little bold and increase the text size of the legends in the figures. As they are a little small compared to the overall text. I feel we have to concentrate a little more to read them.
Also, increase the size of the figures.
Author Response
We really thank the reviewer his/her dedication during the revision of our work. Please find attached the response letter addressing all the comments and suggestions.

Reviewer 2 Report
The article presents a very interesting work from the point of view of computer network simulation. The work combines the simulation of network protocols with the simulation of physical phenomena occurring during transmission in wireless VLC networks. The authors analyze the current standards and - as it seems - with great care reproduce them in the simulation.
Nevertheless, there are elements in the article that raise doubts, sometimes large.
What is important, the simulator has not been validated in the real environment. Even a simple physical configuration with one transmitter, one receiver and different reflective surfaces would make it possible to see that the results achieved in the real world and in the simulation model are similar enough to make the simulator credible.
The article is a reliable report documenting the work done, but the publication serves the purpose of making the results available to members of the community. Meanwhile, the paper contains very few useful tips for potential users (including the reviewer), e.g. how to install the simulator (assuming that the OMNET++ environment is known to the reader), how to describe non-networking elements, especially the room, or whether the simulator described integrates with a higher level framework such as INET. A fair conclusion to the work should be to report it to the creators of OMNET++ and make it available on the simulator's website in the Models and Tools tab, or at least provide a download link to the source code, a simple instruction and one or two examples.
Of lesser importance is the a priori adoption of certain values, such as 200 rays for MRCT modelling. There has not even been a simplified analysis indicating that such a number is sufficient, or an estimate of the error made by limiting MRCT modelling to such a number. On the other hand, the number of simulated rays has consequences in terms of performance, it can be assumed that MRCT is the most time-consuming element of simulation. It is a pity that the authors did not add any report on the speed of the simulation (on a typical workstation).
Author Response
We really thank the reviewer for his/her dedication during the revision process of our work. Please find attached the response letter.

Reviewer 3 Report
This paper presents system-level simulator that considers jointly a channel propagation model and the MAC mechanisms to have a realistic description of the network, even in situations where the emitted signal is heavily affected by reflections in any close surface or obstacle. The simulator was run 15 times in different scenarios to work out network parameters, including throughput, delivery probability and node active time and delivery time and so on. However, there are some questions which are necessary to be raised related to some aspects of the paper for the further improvement:
- According to IEEE 802.15.7r1 standard, the full-name of CSMA/CA is carrier sense multiple access with collision avoidance and is not carrier sense medium access with carrier avoidance which mentioned in your manuscript (paper 1, line 6 & paper 2, line 62). Please double check this abbreviation.
- It is necessary to model VLC MAC layer performances accurately, as you mentioned in your paper. How to evaluate the accuracy of proposed simulation?
- The simulator was run 15 times to obtain statistically results in your paper. However, I believe that the statistical sample should be enormous enough to ensure the credibility of simulation results.
- The delivery time (14 ms) which is related to 50 slots buffer is measured. How to time the slot buffer and slot window length ? Are these times related to your hardware environment which run your simulation ?
Author Response
We thank the reviewer for his/her dedication during the revision of our work. Please find attached the response letter.

Round 2
Reviewer 2 Report
The revised version of the paper, together with the enclosed clarifications in the cover letter, largely dispels our previous concerns. It is still a pity that the authors were unable to verify the simulator in the real world, but the explanation is acceptable. The analysis of the number of rays in the MCRT method is entirely sufficient.
However, we still believe that the simulator code should be available to every reader of the article (including reviewers), along with the paper, not after publishing. It is the code, and not the results of sample simulations, that is the actual result of the authors' research. Without access to it, the evaluation of the work is not complete.
Author Response
We really appreciate the thorough revision process that our work has received. Please find attached the response letter.

Reviewer 3 Report
In their revised version of the manuscript the authors have satisfactorily addressed the concerns by myself and the other reviewer. The paper is now ready for publication in MDPI Sensors in my opinion.
Author Response
We really thank the thorough revision process that our work has received. Please find attached the response letter.
